# Avian Pathogenic *Escherichia coli* through Pfs Affects the Tran-Scription of Membrane Proteins to Resist β-Lactam Antibiotics

**DOI:** 10.3390/vetsci9030098

**Published:** 2022-02-23

**Authors:** Jiangang Hu, Chuanyan Che, Wei Jiang, Zhaoguo Chen, Jian Tu, Xiangan Han, Kezong Qi

**Affiliations:** 1Anhui Province Key Laboratory of Veterinary Pathobiology and Disease Control, Anhui Agricultural University, Hefei 230036, China; vethjg@163.com (J.H.); tujian1980@126.com (J.T.); 2Shanghai Veterinary Research Institute, The Chinese Academy of Agricultural Sciences (CAAS), 518 Ziyue Road, Shanghai 200241, China; jiangwei@shvri.ac.cn (W.J.); zhaoguochen@shvri.ac.cn (Z.C.); 3College of Animal Science, Chuzhou 233100, China; ccyan1980@163.com

**Keywords:** Avian pathogenic *Escherichia coli*, *pfs*, MIC, β-lactam antibiotics, porin protein

## Abstract

Avian pathogenic *Escherichia coli* (APEC) is a causative agent of colibacillosis, one of the principal causes of morbidity and mortality in poultry worldwide. Nowadays, antibiotics are mainly used to prevent and control poultry colibacillosis, but the situation of drug resistance is serious. 5′-methylthioadenosine/S-adenosylhomocysteine nucleosidase (Pfs) is involved in methylation reactions, polyamine synthesis, vitamin synthesis, and quorum sensing (QS) pathways. In this study, compared with the APEC wild-type strain DE17, the *pfs* deletion strain DE17Δpfs was more susceptible to β-lactam antibiotics (amoxicillin, ceftazidime, cefuroxime) by drug sensitivity test and minimum inhibitory concentration (MIC), and the MIC of the DE17Δpfs was half that of the DE17. Quorum sensing signal molecule AI-2 is involved in antibiotic resistance. In the case of pfs inactivation, the DE17Δpfs cannot synthesize AI-2, so it is necessary to add AI-2 to study whether it affects APEC resistance. When the exogenous AI-2 was added, the MIC of all APEC did not change. Transcriptome sequencing indicated that the transcription levels of a lot of outer membrane protein genes and metabolic genes had changed due to the deletion of *pfs*. Moreover, the transcription levels of the efflux pump gene *tolC* and penicillin binding protein (*fstI* and *mrcA*) were significantly reduced (*p* < 0.05), while the transcription levels of the porin protein genes (*ompF*, *ompC*, and *ompD*) were significantly increased (*p* < 0.05). In addition, it was also found that the outer membrane permeability of the DE17Δpfs was significantly increased (*p* < 0.05). The results indicated that *pfs* does not affect APEC strain DE17 resistance to β-lactam antibiotics through AI-2, but *pfs* affects the sensitivity of APEC to β-lactam antibiotics by affecting antibiotic-related genes. This study can provide a reference for screening new drug targets.

## 1. Introduction

Poultry is one of the most widespread types of meat consumed worldwide. Poultry flocks are often raised under intensive conditions, using large amounts of antimicrobial agents to prevent and treat diseases, as well as promote growth. Antimicrobial resistant poultry pathogens may cause treatment failures, economic losses, and are also the source of resistant bacteria and genes (including zoonotic bacteria) that may pose a risk to human health [1]. Avian pathogenic *Escherichia coli* (APEC) causes avian colibacillosis, which results in serious losses to the poultry industry around the world [2,3]. APEC serotypes are complex and may have drug resistance, which severely restricts infection prevention measures [4]. Quorum sensing (QS) is mediated by diffusible small molecules, called autoinducers (AIs), which are synthesized in cells to coordinate group behavior, and QS systems mediate bacterial resistance [5,6,7]. Through AI-2 receptors, AI-2 is considered to be the most common bacterial communication signal, and it is reported to play an important role in regulating many bacterial behaviors (including antibiotic resistance and biofilm formation) through AI-2 receptors [8]. AI-2 is synthesized by the S-ribosylhomocysteine lyase (LuxS) and S-adenosylhomocysteine nucleosidase (Pfs) [9,10]. The production of AI-2 involves the activated methyl cycle (AMC) in bacteria [11]. Pfs is involved in methylation reaction, polyamine synthesis, vitamin synthesis, and QS pathway [12]. AI-2 biosynthesis requires Pfs. Pfs catalyzes two reactions in bacterial cells: the formation of S-ribosylhomocysteine (SRH) from S-adenosylhomocysteine (SAH) to release adenine and the production of 5′-methylthioribose (MTR) from 5′-methylthioadenosine (MTA), while also releasing adenine [13]. Both SAH and MTA are effective inhibitors of the S-adenosylmethionine (SAM) required reaction, and the accumulation of these metabolites is avoided through the activity of Pfs [14]. The lack of pfs in *E. coli* can cause serious growth defects [15]. Previous study has shown that a purified Pfs enzyme is both required and sufficient for AI-2 production in vitro, using SAH as a substrate [16]. However, the sensitivity of *E. coli* to antibiotics is related to AI-2. When the concentration of autoinducers reaches a threshold, population-wide changes in gene expression are triggered, resulting in the modulation of a variety of physiological characteristics, and changes in sensitivity to antibiotics [17,18]. The study has confirmed that exogenous AI-2 increases the resistance of extended-spectrum β-lactamase-positive *E. coli* to β-lactam antibiotics by up-regulating the expression of TEM-type β-lactamase [19]. As an important part of the S-adenosylmethionine (SAM) cycle, in addition to participating in AI-2, Pfs also has many functions in a wide range of metabolic reactions [20]. In previous studies, the effect of Pfs on APEC growth rate, AI-2 activity and motility, and the pathogenicity of APEC decreased after Pfs deletion [21]. Whether *pfs* affects the sensitivity to antibiotics by affecting the synthesis of AI-2 has not been reported. In addition, Pfs has many functions, and it is not reported whether they are involved in bacterial resistance. Therefore, this study wants to study whether *pfs* affects the sensitivity of antibiotics through AI-2. This will help screen out important targets for antibacterial drugs.

## 2. Materials and Methods

### 2.1. Bacterial Strains and Culture Conditions

The APEC strain DE17 was isolated from a sick duck with septicemia and neurological signs [7]. In a previous study, the *pfs* gene deletion strain (DE17Δpfs) and the *pfs* gene complementation strain (c-pfs) were constructed [21]. The bacteria were grown in Luria-Bertani (LB) broth and Mueller-Hinton (MH) broth (Oxoid). The overnight culture was transferred to fresh MH medium, cultivated to an optical density at 600 nm of approximately 1, and then diluted, in fresh MH broth, to an optical density of approximately 0.03 at 600 nm for subsequent experiments.

### 2.2. Disc Diffusion

The APEC strains DE17, DE17Δpfs and c-pfs were tested for susceptibility against 14 antimicrobial agents using the disc-diffusion method in agar MH, as described before [22]. The following antibiotics were used: ampicillin (10 μg), amoxicillin (10 μg), ceftazidime (30 μg), chloramphenicol (30 μg), gentamicin (10 μg), rifampicin (5 μg), tetracycline (30 μg), clindamycin (2 μg), norfloxacin (10 μg), ciprofloxacin (5 μg), doxycycline (30 μg), ciprofloxacin (5 μg), amikacin (30 μg), erythromycin (15 μg). The *E. coli* ATCC 25922 was used as a quality-control strain. The sensitivity of APEC strains to antibiotics was tested via the agar diffusion method, using MH agar and commercially available discs (Oxoid), according to Clinical and Laboratory Standards Institute (CLSI) guidelines [23].

### 2.3. Determination of MIC

The MIC were determined as described previously [24]. The bacteria were cultured to an OD_600_ value of 1. The OD_600_ was adjusted to 0.5 with MH. Antibiotics are serially diluted two-fold in MH medium to make the ampicillin concentration range from 6.354–6250 μg/mL, the amoxicillin concentration from 0.049–500 μg/mL, the ceftazidime concentration from 1.1465–375 μg/mL, and the cefuroxime concentration from 0.244–250 μg/mL. The 96-well plates were prepared by adding 100 μL of MH broth to each well of the 96-well plate, and adding 100 μL of each of the four antibiotic solutions (ampicillin, amoxicillin, cefuroxime, and ceftazidime) to the first well. Then, 100 µL of each of the serial dilutions were transferred into 11 consecutive wells. Next, 95 µL of MH and 5 µL of inoculum were added to each well of the 96-well plate. The last well was used as a negative control, which contained 195 µL of MH without antibiotic and 5 µL of inoculum. The final volume in each well was 200 µL. The 96-well plates were incubated at 37 °C for 20–24 h. In addition, the molecule (S)-4,5-dihydroxy-2,3-pentanedione (DPD) is produced by different species of bacteria and is the precursor of the signaling molecule AI-2. DPD can be used as exogenous AI-2, and different concentrations of DPD (1 mmol/L, 5 mmol/L, and 10 mmol/L) were added to the culture medium to test whether AI-2 would affect the MIC of antibiotics against APEC.

### 2.4. Differential Expression Analysis

The transcriptional levels of DE17 and DE17Δpfs cells were determined as described with some modification [25]. When the OD600 was 1.0, APEC cells were collected by centrifugation at 1800× *g* for 5 min. The cells were washed three times with PBS. According to the manufacturer’s protocol, the total RNA was extracted from the bacterial cell by the Trizol RNA isolation reagent (Invitrogen, Carlsbad, CA, USA). A Nanodrop spectrophotometer was used to evaluate the concentration and quality of RNA through the OD260/OD280 ratio. Then, the illumina HiSe-qTM2500/MiseqTM sequencing platform was used to construct the sequence libraries of DE17 and DE17Δpfs. The main differentially expressed genes (DEGs) were analyzed using gene ontology (GO) enrichment analysis. The Kyoto Encyclopedia of Genes and Genomes (KEGG) database was used to analyze the pathway analysis of DEGs. The Benjamini and Hochberg approaches were used to adjust the *p*-values, and *p* < 0.05 was set as the threshold for significant difference. 

### 2.5. Outer Membrane Permeabilization Assays

The permeability of the bacterial outer membrane was tested using the 1-N-phenylnaphthylamine (NPN) probe method, as described, with some modification [26]. The overnight cultured DE17, Δpfs, and c-pfs were transferred to MH liquid medium and cultivated to an OD_600_ value of 1.2, centrifuged at 1800× *g* for 5 min to collect the bacteria, washed with PBS 3 times, and the value of OD_600_ was adjusted to 0.5. Next, 1.92 mL bacterial solution of DE17, Δpfs, and c-pfs, respectively, were added into a quartz cuvette, and then 80 μL solution of NPN (1 mmol/L) was added to the mix. The fluorescence (the excitation wavelength of 350 nm and emission wavelength of 420 nm) was recorded with an F-4500 FL spectrophotometer (Hitachi, Ltd., Chiyoda, Japan). The value was recorded every 2 min until the number no longer changed, and the value was the highest fluorescence absorption value of the strain. The experiment was repeated three times. 

### 2.6. Detection of Transcription Level by Quantitative Real-Time PCR

The transcription level of the antibiotic resistance related genes was tested by a quantitative real-time PCR (qPCR) experiment. Those antibiotic resistance related genes include the selected efflux pump-encoding gene (*tolC*), and porin protein-encoding genes (*ompC*, *ompD*, and *ompF*), and penicillin-binding protein genes (*mrcA* and *ftsI*). The primers for the target genes and the internal control gene *dnaE* were shown in Table 1. Briefly, the bacteria were grown in MH at 37 °C to mid-log phase (OD_600_ = 1.0), and total RNA was extracted by TRIzol reagent. DNaseI (TaKaRa, Dalian, China) was used to remove DNA from total RNA. The reverse transcription was performed using the PrimeScript 1 st Strand cDNA synthesis kit (TaKaRa, Dalian, China). The qPCR experiment was performed using the SYBR green PCR mix (TaKaRa, Dalian, China). The target genes were examined three times, and relative changes in gene expression levels were assessed using the 2^−∆∆Ct^ method [25]. 

### 2.7. Statistical Analysis

The statistical analyses were conducted by SPSS V19.0 software (SPSS, Chicago, IL, USA). Statistically significant differences between mean values were identified using the student′s *t*-test. A *p*-value of less than 0.05 was considered significant. 

## 3. Results 

### 3.1. Disc Diffusion

The results showed that all APEC strains—DE17, DE17Δpfs, and c-pfs—were resistant to clindamycin, erythromycin, tetracycline, and amikacin. It is susceptible to the other ten antibiotics. APEC wild-type strain DE17, deletion strain DE17Δpfs, and complement strain c-pfs were all susceptible to the β-lactam antibiotics (ampicillin, amoxicillin, and ceftazidime). Interestingly, after *pfs* gene deletion, the inhibition zone of the deletion strain DE17Δpfs was larger than that of the wild-type strain and complement strain. The inhibition zone of the two antibiotics (amoxicillin and cefuroxime) against the DE17 was 15 mm, while the inhibition zone for the DE17Δpfs increased to 20 mm, indicating that *pfs* gene deletion was more susceptible to the cephalosporin antibiotics (Figure 1).

### 3.2. MIC Result

The MIC of APEC was determined using four β-lactam antibiotics (ampicillin, amoxicillin, cefuroxime, and ceftazidime) were selected to determine the MIC of APEC. The results are shown in Table 2. The MIC of ampicillin against the DE17, DE17Δpfs, and c-pfs was same (12.707 μg/mL). The MIC of amoxicillin against the wild-type strain DE17 and complement strain c-pfs was 7.813 μg/mL, and the MIC for deletion strain DE17Δpfs was 3.906 μg/mL. The MIC of cefuroxime against DE17 and c-pfs was 3.906 μg/mL, and the MIC for deletion strain DE17Δpfs was 1.953 μg/mL. The MIC of ceftazidime against DE17 and c-pfs was 5.859 μg/mL, and the MIC for deletion strain DE17Δpfs was 2.930 μg/mL. DPD is an exogenous AI-2, and different concentrations of DPD are added to the culture medium to affect antibiotic resistance. It was found that there was no change in the MIC of these antibiotics after the addition of DPD to the culture medium. The results showed that the sensitivity of APEC to amoxicillin, cefuroxime, and ceftazidime increased after the deletion of the *pfs* gene. It is possible that *pfs* does not affect the sensitivity of antibiotics through the AI-2 pathway.

### 3.3. Determination of Differentially Expressed Genes and Pathway Analysis

The transcriptome analysis showed that, compared with the wild-type strain DE17, 742 genes in DE17Δpfs were up-regulated, and 364 genes were down-regulated (Figure 2) (See Appendix A). The up-regulated and down-regulated genes were analyzed across three categories: biological processes, cellular components, and molecular functions (Figure 3A). The GO enrichment analysis was associated with the catalytic activity, with 39 genes. Some terms could influence the membrane and the membrane part, and the total number of genes affected in these two parts is 65. Furthermore, KEGG pathway analysis showed that metabolic genes were most affected by *pfs* deletion, including 346 global and overview maps genes (Figure 3B). Meanwhile, it was also found that the transcription level of 127 genes related to membrane transport has changed. In general, the GO and KEGG analysis showed that Pfs had a great impact on the membrane of APEC.

### 3.4. Permeabilization of the Outer Membrane

NPN is a highly susceptible fluorescent probe molecule, which is widely used to determine the permeability of the outer membrane of Gram-negative bacteria. The relative fluorescence absorption values of DE17, Δpfs, and c-pfs were detected using NPN at a final concentration of 30 μmol/L (Figure 4). The fluorescence absorption value of the DE17Δpfs strain was two times that of the DE17 (*p* < 0.05), and the fluorescence absorption value of the DE17Δpfs strain was 1.6 times that of the complementation strain (*p* < 0.05). The results indicated that the outer membrane permeability of APEC was enhanced when the *pfs* gene was deleted.

### 3.5. Gene Transcription Levels Associated with β-Lactam Antibiotic Resistance

According to the analysis of the transcriptome, we used qPCR to detect the transcription level of β-lactam resistance related genes. The qPCR results revealed changes in the mRNA levels of β-lactam resistance-related genes in APEC following *pfs* deletion (Figure 5). Compared with the wild-type strain DE17, the selected efflux pump-encoding gene *tolC* and penicillin-binding protein genes (*fstI* and *mrcA*) transcription levels were significantly reduced after *pfs* deletion (*p* < 0.05). However, the transcription level of 3 porin protein-encoding genes (*ompF*, *ompC*, and *ompD*) increased significantly (*p* < 0.05). Compared with the wild strain, the deletion strain was up-regulated 2.5-fold, 1.5-fold, and 3.0-fold, respectively. The transcription levels of these genes were restored in complementary strains. 

## 4. Discussion

In this study, it was found that, after *pfs* deletion, APEC was more susceptible to β-lactam antibiotics, especially cephalosporins antibiotics. Although the strain DE17Δpfs does not produce the signal molecule AI-2, and AI-2 has been reported to be associated with β -lactam antibiotic sensitivity, in this study, the addition of exogenous AI-2 did not make APEC resistant, so the sensitivity of strain DE17Δpfs to β-lactam antibiotics was not associated with the absence of AI-2 [19]. All β-lactam antibiotics have a bactericidal effect by interfering with the biosynthesis of peptidoglycan, a component of the bacterial cell wall. The cell wall of bacteria is a complex structure, composed of tightly cross-linked peptidoglycan network. The cell wall can maintain the shape of the cell. The periplasmic/extra-cytoplasmic target of penicillin are a family of enzymes with highly conserved catalytic activity, involved in the final process of bacterial cell wall biosynthesis: cross-linking of structural polymer peptidoglycan (PG) [27]. The penicillin-binding proteins (PBPs) are essential enzymes that catalyze this crosslinking step. The PBPs mistakenly use penicillin as a substrate for cell wall synthesis, and transpeptidase (or carboxypeptidase) is acylated [28]. Acylated PBPs are unable to hydrolyze β-lactam, causing the subsequent steps of cell wall synthesis to be hampered, and the autolysis of cell wall degradation (autolysis) enzymes to continue, causing bacterial cells to become permeable to water, quickly absorb liquid, and finally lyse [28,29]. The four high molecular weight PBPs of *E. coli* (PBPs 1a, 1b, 2 and 3) are responsible for the synthesis and assembly of the peptidoglycan sacculus that forms the rigid bacterial cell wall [30]. In this study, it was found that the transcription level of penicillin binding proteins FstI (PBPs 3) and MrcA (PBPs 1a) decreased in deletion strain DE17Δpfs, which affected the synthesis of the cell wall, thereby changing the permeability, making it susceptible to β-lactam antibiotics. According to other studies, ceftazidime preferentially inhibits FtsI, a cytoplasmic membrane protein that acts as a DD-transpeptidase in the formation of the septum at the murein sacculus, and is the main lethal target of β-lactam antibiotics [31,32]. In fact, FtsI overproduction alone can increase ceftazidime MICs by four- to eight-fold [33]. This is consistent with the results of this study.

The strategy of resistance to β-lactams, on the other hand, is to reduce drug concentration by increasing efflux, or in the case of gram-negative bacteria, by reducing penetration into the periplasm [34]. The outer membrane plays an important role in this process. The outer membrane of gram-negative bacteria protects them from external toxic compounds. Porins are channel forming proteins that allow small hydrophilic solutes to diffuse through the membrane’s barrier [35,36]. Bacteria’s resistance to various antibiotics is due to the reduced permeability of the outer membrane [37]. The natural OmpF porin of *E. coli* is a trimer. The three-dimensional structure shows a monomeric β-barrel composed of 16 anti-parallel β-strands containing pores [38]. Studies have shown that OmpF-deficient mutants were resistant to several antibiotics, including β-lactam, which indicates that OmpF was the main pathway for β-lactam antibiotics to penetrate the outer membrane [13]. Most β-lactam antibiotics are transported via non-specific porins, especially OmpF and OmpC, and it is important that the outer membrane porins are the key to β-lactam penetration [39]. The transcription level of the porin protein-encoding genes (*ompC*, *ompD*, and *ompF*) was significantly increased in the strain DE17Δpfs in our study, which could explain the sensitivity to β-lactam antibiotics. When *pfs* is deleted, the expression of outer membrane protein increases, allowing antibiotics to enter the bacteria’s outer membrane and act on the cell wall more easily. Hence, it is easier to inhibit the growth of bacteria. The outer membrane permeability test of bacteria also verified this. The outer membrane permeability of the bacteria also becomes stronger in strain DE17Δpfs. The possible reason is that it is also caused by the increased expression of porin protein. In addition, we also examined the transcription level of the efflux pump gene. The outer membrane protein TolC is an envelope protein found in all Gram-negative bacteria, and it is essential for the excretion of excessive compounds [40,41]. The AcrAB-TolC system can expel some β-lactams with multiple charged groups. A lack of the AcrAB-TolC system will result in reduced efflux of substrates, leading to reduced bacterial resistance [42]. The transcriptional level of tolC was reduced in this study, which may result in a reduced ability of APEC to efflux antibiotics, resulting in increased sensitivity to β-lactam antibiotics. Pfs can achieve β-lactam antibiotic resistance by regulating APEC outer membrane porin gene (*ompF*, *ompC*, *ompD*), penicillin binding protein genes (*fstI* and *mrcA*), and efflux pump gene (*tolC*) (as shown in the Figure 6). The external AI-2 does not participate in this process.

In addition to participating in the synthesis of AI-2, Pfs also has many functions, such as methylation reactions, recovery pathways for adenine, and the synthesis of polyamines and vitamins. Transcriptome data proves that *pfs* affects the expression of many metabolic genes. These metabolic genes may cause metabolic imbalances, which may be the main reason for the significant changes in antibiotic-related genes.

## Figures and Tables

**Figure 1 vetsci-09-00098-f001:**
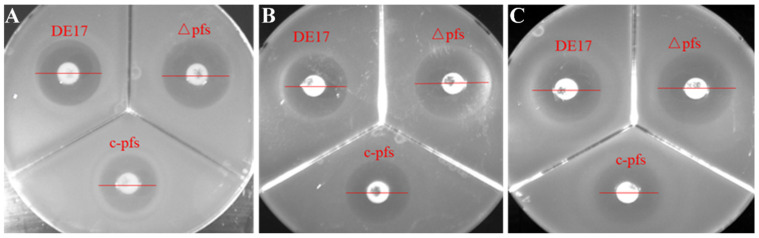
The size of the inhibition zones of different antibiotics against APEC strains. (**A**) Ampicillin, (**B**) amoxicillin, and (**C**) ceftazidime. After *pfs* gene deletion, the inhibition zone of the deletion strain DE17Δpfs was larger than that of wild-type strain DE17 and complement strain c-pfs.

**Figure 2 vetsci-09-00098-f002:**
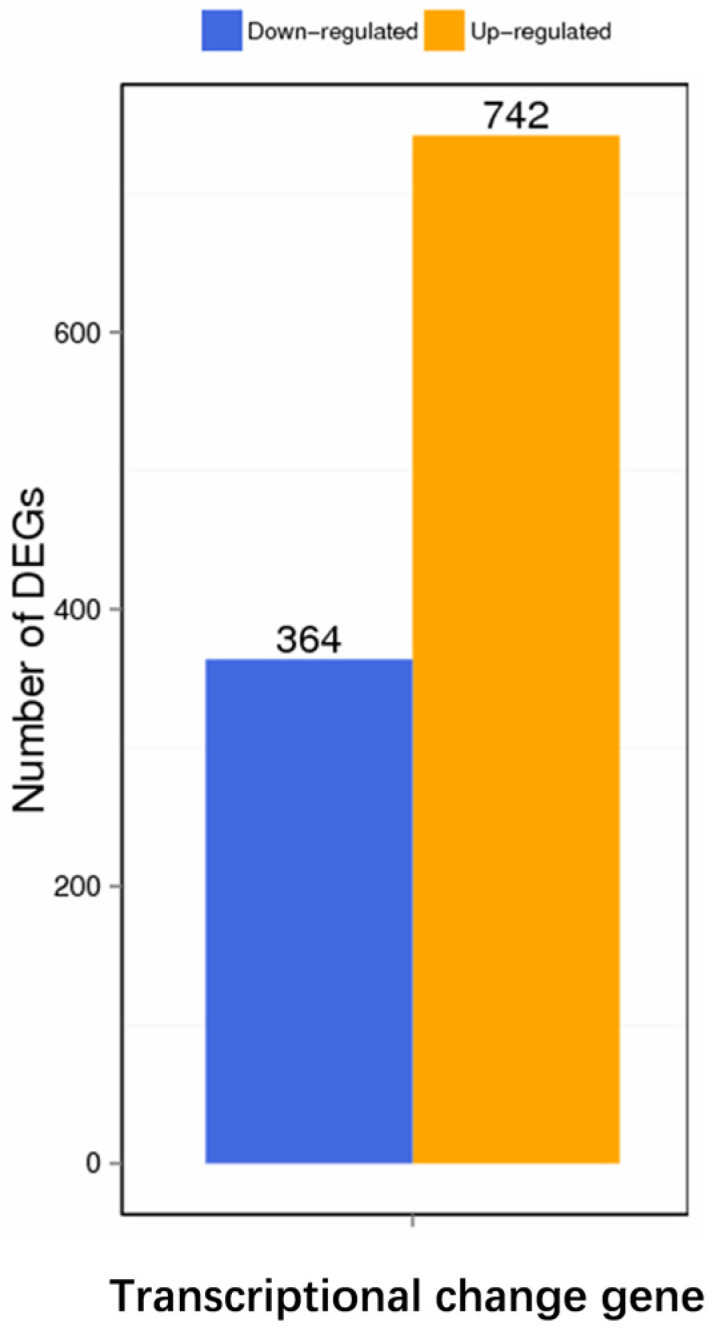
Differential genes expression. Compared with the wild-type strain, there were 742 up-regulated genes and 364 down-regulated genes after *pfs* deletion.

**Figure 3 vetsci-09-00098-f003:**
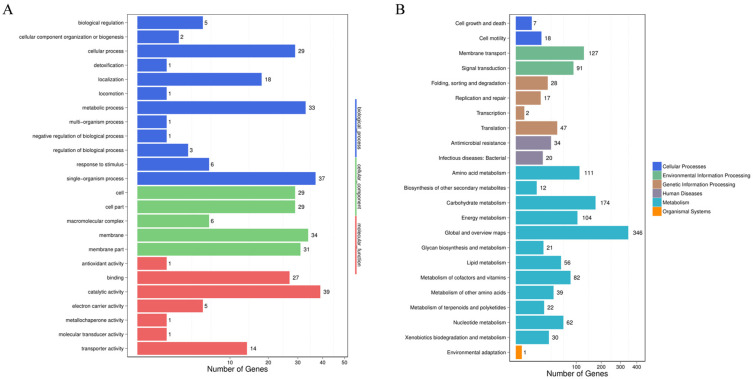
Differential genes analysis. (**A**) Analysis of significant enrichment in GO function of differential genes. The differentially expressed genes directly reflected the number and distribution of differentially expressed genes by GO term enriched. The abscissa is the number of differential genes, and the ordinate is GO terms. There are three categories of GO terms (biological progress, cellular component, and molecular function), which are marked with different colors. (**B**) Analysis of the pathway’s significant enrichment of differential genes. The differentially expressed genes directly reflected the number and distribution of differentially expressed genes through the KEGG public database. The abscissa is the number of differential genes, and the ordinate is the secondary KEGG pathway. The secondary pathways belong to different primary pathways, and different colors are used to indicate the primary pathway category in the figure.

**Figure 4 vetsci-09-00098-f004:**
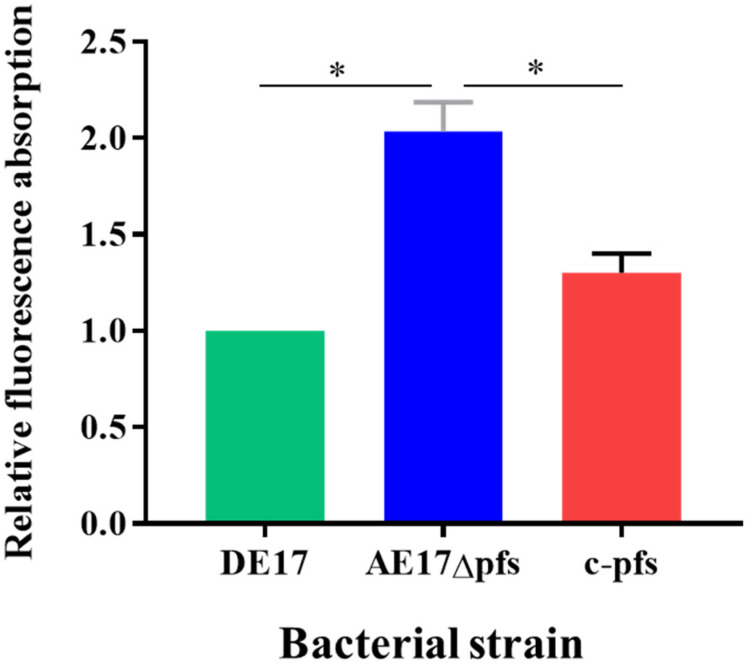
The APEC membrane permeability was determined by measuring the fluorescence resulting from the uptake of 1-N-phenyl-naphthylamine (NPN). The results are presented as the means ± SE of three independent experiments, (ns, not significant; * *p* < 0.05).

**Figure 5 vetsci-09-00098-f005:**
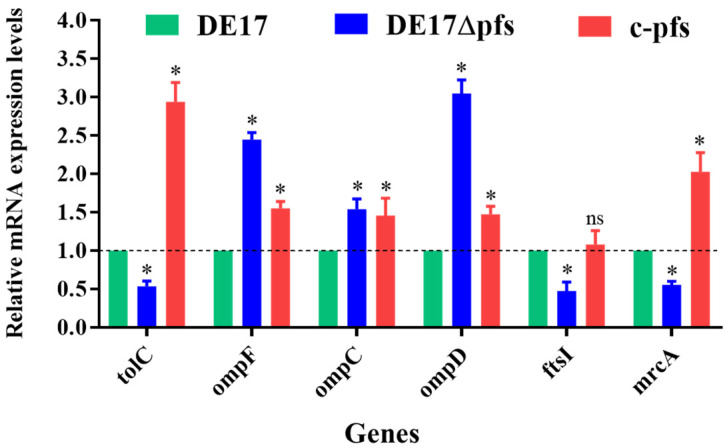
Transcriptional analysis of β-lactam resistance-related genes. The qPCR results showed that, compared with DE17, the mRNA levels of the three genes (*tolC*, *ftsI*, and *mrcA*) in the *pfs* mutant strain were significantly reduced (*p* < 0.05). The mRNA levels of 3 genes (*ompF*, *ompC*, and *ompD*) increased significantly (*p* < 0.05) (ns, not significant; *, *p* < 0.05).

**Figure 6 vetsci-09-00098-f006:**
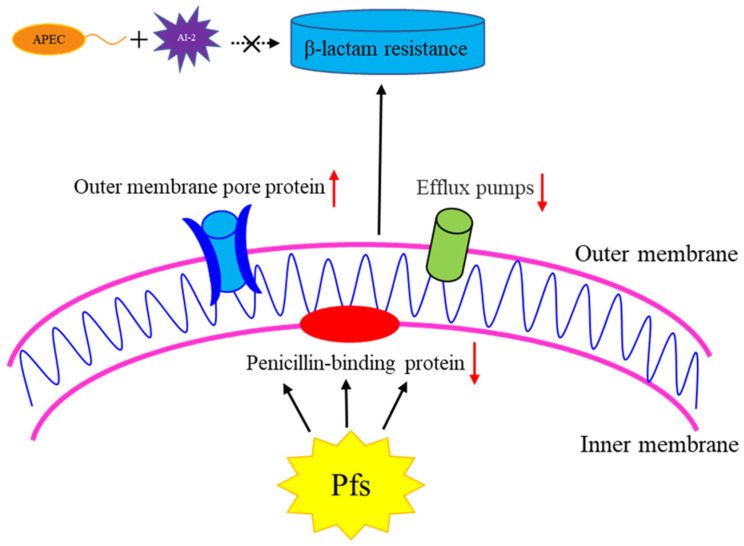
The role of Pfs in β-lactam resistance in APEC strain DE17. By regulating the outer membrane pore protein gene, penicillin binding protein gene, and efflux pump gene in APEC, the Pfs can achieve resistance to β-lactam antibiotics. The upward red arrow indicates that the gene transcription level is up-regulated, and the red arrow downward indicates that the gene transcription level is down-regulated. Exogenous AI-2 does not participate in this process.

**Table 1 vetsci-09-00098-t001:** RT-qPCR primers used in this study.

Primer Name	Sequence (5′–3′)	Description	Product Size (bp)	Source
*tolC*-RT-F	CTGAAAGAAGCCGAAAAACG	Partial DNA sequence of TolC	207	This study
*tolC*-RT-R	CTGGCCCATATTGCTATCGT			This study
*ompC*-RT-F	ACGCAGAGCTTTACGACCAT	Partial DNA sequence of OmpC	178	This study
*ompC*-RT-R	CTTTATGCAGCAGCGTGGTA			This study
*ompD*-RT-F	GTCCTGATCACCCCAAACAC	Partial DNA sequence of OmpD	184	This study
*ompD*-RT-R	CATCTATCTGGCCACCACCT			This study
*ompF*-RT-F	TATTTAAGACCCGCGAATGC	Partial DNA sequence of OmpF	161	This study
*ompF*-RT-R	GACATGACCTATGCCCGTCT			This study
*fstI*-RT-F	TGTGCGTAAAGACCGCTATG	Partial DNA sequence of FstI	183	This study
*fstI*-RT-R	GTGTTGACATCCACCAGCAC			This study
*mcrA*-RT-F	ACCACGTTTTTCGACTGACC	Partial DNA sequence of McrA	197	This study
*mcrA*-RT-R	GTGGGTTCCAACATCAAACC			This study
*dnaE*-RT-F	TGGCCTACGCGTTAAAAATC	Partial DNA sequence of the internal control DnaE	157	This study
*dnaE*-RT-R	TACATGTCCGCTACGTGCTC			This study

**Table 2 vetsci-09-00098-t002:** The MIC values of antibiotics against APEC tested with micro-dilution assay.

Antibiotic	Antimicrobial MIC		
	DE17	DE17Δpfs	c-pfs
Ampicillin	12.707	12.707	12.707
Ampicillin + 1 mM/L DPD	12.707	12.707	12.707
Ampicillin + 5 mM/L DPD	12.707	12.707	12.707
Ampicillin + 10 mM/L DPD	12.707	12.707	12.707
Amoxicillin	7.813	3.906	7.813
Amoxicillin + 1 mM/L DPD	7.813	3.906	7.813
Amoxicillin + 5 mM/L DPD	7.813	3.906	7.813
Amoxicillin + 10 mM/L DPD	7.813	3.906	7.813
Cefuroxime	3.906	1.953	3.906
Cefuroxime + 1 mM/L DPD	3.906	1.953	3.906
Cefuroxime + 5 mM/L DPD	3.906	1.953	3.906
Cefuroxime + 10 mM/L DPD	3.906	1.953	3.906
Ceftazidime	5.859	2.930	5.859
Ceftazidime + 1 mM/L DPD	5.859	2.930	5.859
Ceftazidime + 5 mM/L DPD	5.859	2.930	5.859
Ceftazidime + 10 mM/L DPD	5.859	2.930	5.859

## Data Availability

The raw data supporting the conclusions of this article will be made available by the authors, without undue reservation, to any qualified researcher.

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
