# Peer review of "Avian Pathogenic Escherichia coli through Pfs Affects the Tran-Scription of Membrane Proteins to Resist β-Lactam Antibiotics"

_vetsci, 2022, doi:10.3390/vetsci9030098_

Round 1

Reviewer 1 Report

Line 40: explain APEC

Line 42: explain QS

Line 52: explain Pfs   Line 245: explain ExPEC

Author Response

Point 1: Line 40: explain APEC

Response 1: Thank you for your good suggestions, we have explained APEC and marked in yellow in the revised manuscript (Line 44).

Point 2: Line 42: explain QS

Response 2: Thank you for your good suggestions, we have explained QS and marked in yellow in the revised manuscript (Line 47).

Point 3: Line 52: explain Pfs

Response 3: Thank you for your good suggestions, we have explained Pfs and marked in yellow in the revised manuscript (Lines 52-53).

Point 4: Line 245: explain ExPEC

Response 4: Thank you for your good suggestions, as suggested by reviewers, we have deleted the contents related to ExPEC.

The revised manuscript is uploaded as an attachment.

Reviewer 2 Report

The manuscript (ref. no.:Vet Sci-1557059) reported a study about 5’-methylthioadenosine/S-adenosylhomocysteine nucleosidase (Pfs) affects the transcription of membrane proteins to resist  β-lactam antibiotics in avian pathogenic E. coli. This study constructed the deletion strain and complement strain of pfs gene and studied the molecular mechanism of pfs gene in mediating β-lactam antibiotics. However, there are flaws with the study design which do not allow sound conclusions on some causal relationships stated in the manuscript. A comprehensive data interpretation is lack. The data analysis is not deep and there are many conceptual error and grammatical error. Thus, a major revision is required before the manuscript can be considered for publication. The major concerns are provided below:

  1. Line 88: “using the in agar MH”
  2. Please add the related references of antibiotic susceptibility testing.
  3. Section 2.2 and 2.3 were repeated. What is different between antibiotic susceptibility testing and determination of MIC? Please check the title of section 2.3.
  4. It is unreasonable to adjust the OD to 0.5 to determine the MIC. Please check it.
  5. Line 111: mM/L changed as mmol/L or mM.
  6. Line 148: “First, Briefly, ...” is not correct. Please revise it.
  7. Line 177: “Antibacterial activity of MIC”. Please check it.
  8. Results Line 192-203. Add the differential expression genes in supporting information. How do you identify these differential expression genes? Pathway analysis found a total of 111 genes were involved in amino acid metabolism. Why Why no further verification?
  9. Figure 2: Please add the title of the abscissa of the figure.
  10. Figure 4 and Figure 5: The title of the abscissa and ordinate of this figure needs to be modified.
  11. Line 235: What is the fold change of the transcription level of the three porin propein-encoding genes?
  12. Discussion: Line 244-253, it is not related with the main content of this study. Please delete it.
  13. Why not detect the expression of some known β-lactam antibiotic resistance genes (such as ampC, ESBL)?  
  14. Line 326: “Metabolic imbalances”? How to explain?

Author Response

Point 1:  Line 88: “using the in agar MH”

Response 1: Thanks for your good suggestion, we had modified the sentences in revised manuscript.  The modified place has been highlighted (Line 88).

Point 2: Please add the related references of antibiotic susceptibility testing.

Response 2: Thank you for your good suggestions, we have added references in the revised manuscript (Line 88). Please see the following the papers.

[1] Oliveira, A.M.S.; Baraúna, R.A.; Marcon, D.J.; Lago, L.A.B.; Silva, A.; Lusio, J.; Tavares, R.D.S.; Tacão, M.; Henriques, I.; Schneider, M.P.C. Occurrence, antibiotic-resistance and virulence of e. Coli strains isolated from mangrove oysters (crassostrea gasar) farmed in estuaries of amazonia. Mar Pollut Bull 2020, 157, 111302.

Point 3: Section 2.2 and 2.3 were repeated. What is different between antibiotic susceptibility testing and determination of MIC? Please check the title of section 2.3.

Response 3: Thank you for your good suggestions, we have changed the section 2.2 to “Disc diffusion” (Line 86). Disc diffusion and MIC are used to study antimicrobial activity, we first screened the antibiotics that were sensitive to pfs deletion by disc diffusion, and then further determined the sensitivity to β-lactam antibiotics after pfs deletion by MIC. This gives the fold change in antibiotic susceptibility.

Point 4: It is unreasonable to adjust the OD to 0.5 to determine the MIC. Please check it.

Response 4: The method of MIC was performed with reference to previous papers, the bacteria cultured to an OD600 value of 1 adjust the OD600 to 0.5 with MH. Please see the following the papers.

[1] Lagha, R.; Ben Abdallah, F.; Al-Sarhan, B.O.; Al-Sodany, Y. Antibacterial and biofilm inhibitory activity of medicinal plant essential oils against isolated from uti patients. Molecules 2019, 24.

Point 5: Line 111: mM/L changed as mmol/L or mM.

Response 5: Thank you for your good suggestions, we have changed “mM/L” as “mmol/L” and marked in yellow in the revised manuscript (Line 111).

Point 6: Line 148: “First, Briefly, ...” is not correct. Please revise it.

Response 6: Thanks for your good suggestion, we had modified the sentences in revised manuscript. The modified place has been highlighted (Line 143).

Point 7: Line 177: “Antibacterial activity of MIC”. Please check it.

Response 7: Thank you for your good suggestions, we have changed “Antibacterial activity of MIC” as “MIC result”. The modified place has been highlighted (Line 171).

Point 8: Results Line 192-203. Add the differential expression genes in supporting information. How do you identify these differential expression genes? Pathway analysis found a total of 111 genes were involved in amino acid metabolism. Why Why no further verification?

Response 8: Thank you for your good suggestions, we have added information on differential expression genes in the revised manuscript (Line 189). These differential expression genes were screened by transcriptome sequencing analysis. We did not analyze the reasons for metabolic genes, mainly because there were many changes in outer membrane-related genes, and the main resistance mechanisms of β-lactam antibiotics were closely related to outer membrane porins, efflux pumps, and penicillin-binding proteins. Therefore, there is no further analysis of metabolism-related genes, which can be used as a direction for our future research.

Point 9: Figure 2: Please add the title of the abscissa of the figure.

Response 9: Thank you for your good suggestions, we have added the title of the abscissa of the figure 2 in the revised manuscript (Line 198).

Point 10: Figure 4 and Figure 5: The title of the abscissa and ordinate of this figure needs to be modified.

Response 10: Thank you for your good suggestions, we have modified figures 4 and 5 in the revised manuscript (Line 219 and line 233).

Point 11: Line 235: What is the fold change of the transcription level of the three porin propein-encoding genes?

Response 11: Compared with the wild strain, the transcription level of 3 porin protein-encoding genes (ompF, ompC, ompD) increased significantly (p < 0.05). The deletion strain was up-regulated 2.5-fold, 1.5-fold, and 3.0-fold, respectively. we had modified the sentences in revised manuscript (Lines 230-232).

Point 12: Discussion: Line 244-253, it is not related with the main content of this study. Please delete it.

Response 12: Thank you for your good suggestions, we have deleted it in the revised manuscript.

Point 13: Why not detect the expression of some known β-lactam antibiotic resistance genes (such as ampC, ESBL)?

Response 13: Thank you for your good suggestions. Transcriptome sequencing found that there were many changes in membrane-related genes, and the main drug resistance mechanism of β -lactam antibiotics was closely related to outer membrane pore protein, efferent pump and penicillin binding protein. However, no β-lactam antibiotic resistance genes were screened in transcriptome analysis. Therefore, there is no further analysis of β-lactam antibiotic resistance genes, which can be used as the direction of our future research.

Point 14: Line 326: “Metabolic imbalances”? How to explain?

Response 14: Metabolism changes in the deletion strain compared to the wild strain. Due to the deletion of pfs gene, the most changes in transcription level are in amino acid metabolism genes, so we speculated that changes in membrane genes may be caused by metabolism, which needs to be confirmed in future studies.

The revised manuscript is uploaded as an attachment.

Reviewer 3 Report

The topic of the manuscript is a very interesting research project. The manuscript was written correctly, but I have some minor comments:

-line 89: please correct or explain: "using the in agar MH"

- line 95: Authors should refer to the full name of the standard and the year of its publication (each subsequent edition brings some changes), as well as introduce it to the list of references

- paragraph "Determination of MIC": I do not understand why the authors refer to the methodology described in publication No 22. The evaluation of the MIC value for the antibiotics used in the manuscript is included in the CLSI standard, which is one of the most important reference materials for the evaluation of the MIC value . All guidelines are in the M7-A10 and one of them , among other things, is not to exceed a total volume of 100 µl when determining susceptibility using the micro-dilution method. The authors declare a total volume of 200 µl (line108). So was the study reliable?

Line 11-113: How many replicates have a susceptibility  assessment been performed?

Results: please use "susceptible" rather than "sensitive" for antibiotics

Line 168-172: Since the disk diffusion method is not a fully reliable method and has many critical points, how many replicates have the assay been performed?

Line 231-232: the meaning of this sentence is not very clear, please specify

There was no summary conclusion in the discussion

Author Response

Point 1: please correct or explain: "using the in agar MH"

Response 1: Thanks for your good suggestion, we had modified the sentences in revised manuscript.  The modified place has been highlighted (Line 88).

Point 2: line 95: Authors should refer to the full name of the standard and the year of its publication (each subsequent edition brings some changes), as well as introduce it to the list of references

Response 2: Thank you for your good suggestions, we have added references in the revised manuscript (Line 95). Please see the following the papers.

Ceriotti, F.; Zakowski, J.; Sine, H.; Altaie, S.; Horowitz, G.; Pesce, A.J.; Boyd, J.; Horn, P.; Gard, U.; Horowitz, G. Clinical and laboratory standards institute (clsi). 2012.

Point 3: paragraph "Determination of MIC": I do not understand why the authors refer to the methodology described in publication No 22. The evaluation of the MIC value for the antibiotics used in the manuscript is included in the CLSI standard, which is one of the most important reference materials for the evaluation of the MIC value . All guidelines are in the M7-A10 and one of them , among other things, is not to exceed a total volume of 100 µl when determining susceptibility using the micro-dilution method. The authors declare a total volume of 200 µl (line108). So was the study reliable?

Response 3: Thank you for your good suggestions. The method of MIC is reliable, and the total volume of MIC has exceeded 100 µl in many published papers. Some previous studies have provided us with reference (please see the following three papers).

[1] Hong, L.; Jing, W.; Qing, W.; Anxiang, S.; Mei, X.; Qin, L.; Qiuhui, H. Inhibitory effect of zanthoxylum bungeanum essential oil (zbeo) on escherichia coli and intestinal dysfunction. Food Funct 2017, 8, 1569-1576.

[2] Wandee, S.; Chan, R.; Chiemchaisri, W.; Chiemchaisri, C. Alteration of antibiotic-resistant phenotypes and minimal inhibitory concentration of escherichia coli in pig farming: Comparison between closed and open farming systems. Sci Total Environ 2021, 781, 146743.

[3] Lagha, R.; Ben Abdallah, F.; Al-Sarhan, B.O.; Al-Sodany, Y. Antibacterial and biofilm inhibitory activity of medicinal plant essential oils against isolated from uti patients. Molecules 2019, 24.

Point 4: Line 11-113: How many replicates have a susceptibility assessment been performed?

Response 4: The susceptibility assessment was repeated three times.

Point 5: Results: please use "susceptible" rather than "sensitive" for antibiotics

Response 5: Thanks for your good suggestion, we had modified the sentences in revised manuscript.  The modified place has been highlighted.

Point 6: Line 168-172: Since the disk diffusion method is not a fully reliable method and has many critical points, how many replicates have the assay been performed?

Response 6: The disk diffusion test was repeated for three times, and MIC was tested for further confirmation.

Point 7: Line 231-232: the meaning of this sentence is not very clear, please specify

Response 7: Thank you for your good suggestions, we had modified the sentences in revised manuscript. The modified place has been highlighted (Lines 225-226).

Point 8: There was no summary conclusion in the discussion

Response 8: Thank you for your good suggestions, We have added conclusions to the discussion in the revised manuscript (Lines 293-297)

The revised manuscript is uploaded as an attachment.

Round 2

Reviewer 2 Report

This article is an interesting study and demonstrate the effect of pfs on bacterial drug resistance by RNA-seq, qRT-PCR and outer membrane permeability assay. The authors have revised the manuscript according to my suggestions. I suggest to accept this article after modifying some grammatical errors.

Author Response

Point 1: I suggest to accept this article after modifying some grammatical errors.

Response 1: Thank you for your good suggestions, we have corrected the wrong grammar in revised manuscript. The modified place has been highlighted.

Reviewer 3 Report

I have no any other comments

Author Response

 Thank you for your good suggestions, we have modified some English language and style in revised manuscript. The modified place has been highlighted.
